# A Systematic Review of Sexual Minority Women’s Experiences of Health Care in the UK

**DOI:** 10.3390/ijerph16173032

**Published:** 2019-08-21

**Authors:** Catherine Meads, Ros Hunt, Adam Martin, Justin Varney

**Affiliations:** 1Faculty of Health, Education, Medicine and Social Care, Anglia Ruskin University, Cambridge CB1 1PT, UK; 2Academic Unit of Health Economics, Leeds Institute of Health Sciences, University of Leeds, Leeds LS2 9JT, UK; 3Director of Public Health, Birmingham City Council, Birmingham City Council’10 Woodcock Street, Birmingham B7 4BL, UK

**Keywords:** sexual minority women, SMW, lesbian, bisexual, trans, health inequalities, heterosexism

## Abstract

Sexual minority women (SMW) experience worse health and disproportionate behavioural risks to health than heterosexual women. This mixed-methods systematic review evaluated recent studies on health experiences of UK SMW, published 2010–2018. Analysis was through narrative thematic description and synthesis. Identified were 23,103 citations, 26 studies included, of which 22 provided qualitative and nine quantitative results. SMW had worse health experiences that might impact negatively on access, service uptake and health outcomes. Findings highlighted significant barriers facing SMW, including heteronormative assumptions, perceptions and experiences of negative responses to coming out, ignorance and prejudice from healthcare professionals, and barriers to raising concerns or complaints. Little information was available about bisexual and trans women’s issues. Findings highlighted the need for explicit and consistent education for healthcare professionals on SMW issues, and stronger application of non-discrimination policies in clinical settings.

## 1. Background

Sexual minority women (SMW) include women defining themselves by sexual identity (lesbians, bisexual women), behaviour (women who have sex with women, women who have sex with men and women) or relationship status (women who are married to or cohabit with other women).

Although there is a limited evidence base [1,2,3,4], in general, SMW experience worse mental health [5], worse physical health [6,7] and higher risk factors for physical ill-health [8,9,10,11] than their heterosexual counterparts. Due to lack of outcome-focused research [12], it is unclear whether difficulties with healthcare access are driving worse physical and mental health.

There have been several international systematic reviews on SMW’s experiences of healthcare in specific settings. A systematic review of lesbian disclosure to primary care providers [13] included 30 studies (one from UK). It found that a wide variety of attributes of lesbians, healthcare providers and setting affected disclosure. Safety was important for disclosure as was relevancy, health status, how likely a person was to be out overall, and relationship status. The review highlighted the importance of enquiring about sexual orientation rather than presuming heterosexuality. Socio-demographic factors such as age, ethnicity and education did not have clear links with disclosure. 

A meta-ethnographic systematic review of lesbian’s experiences of childbirth [14] included 13 studies (four from UK). They identified four main themes: encountering and managing overt and covert prejudice, acknowledging the confidence that can be created when professionals present knowledge about lesbian lifestyle and even small gestures of appropriate support, disclosure of sexual orientation being important but risky unless the patient was in charge of the context or situation, and the need for acceptance of the lesbian family by recognising both mothers. 

A systematic review of sexual minority people’s needs and experiences for end of life and palliative care [15] included 12 studies (one from UK); most of the information for women was related to cancer. The evidence consistently showed the need for all of the health professionals involved in end of life care to be better educated to explore sexual preferences of their patients, avoid heterosexist assumptions, and recognise the importance of partners in decision-making. Health professionals also need to recognise the importance of supportive groups where sexual minority people feel safe to reveal their sexuality, feel accepted and be understood by the support group. 

Reasons why sexual minority people may not feel comfortable about revealing their sexual orientation include heteronormativity or overt homophobia. Heteronormativity is the assumption that people are heterosexual. This can result in attitudes and behaviours that exclude people who are not heterosexual (for example assuming a woman of reproductive age who is having regular sexual activity may become pregnant unless contraception is used). Homophobia in a healthcare related setting can manifest as inappropriate refusal to provide care, providing sub-optimal care or inappropriate words or behaviour whilst providing care. 

There have been no recent systematic reviews covering the experiences of SMW in a breadth of settings nor specifically from the UK. This systematic review includes all recent evidence on SMW’s experiences of UK healthcare in a variety of settings. It focuses on UK research only as experience of healthcare is likely to be very different in other countries because of differences in healthcare delivery and different perceptions of homosexuality and bisexuality. This is a mixed-methods systematic review using both qualitative and quantitative methods on the same topic because neither alone can provide the richness of information available. Mixed methods systematic reviews can provide triangulation of results and increased value compared to either method on its own, and increase the relevance of the findings for decision makers [16]. 

## 2. Methods

A protocol for the whole project investigating all aspects of health and experience of healthcare in SMW was registered with the Prospero database (No. CRD42016050299). This part of the project investigated experiences of UK healthcare in any setting by SMW (lesbians, bisexual women, women who have sex with women (WSW) and women who have sex with men and women (WSMW), same sex married or cohabiting women or other non-defined non-heterosexual women). Trans women were included if they also identified as SMW. Self-report or objectively measured health experiences were included, from any published or unpublished research (i.e., grey literature reports available on LGBT organisation websites) dated from 2010 onwards. 

### 2.1. Searches

Searches were conducted in June 2018 and included results from previous searches for related projects. Databases (platforms) searched were CAB abstracts (Ovid), Cinahl (Elsevier), Cochrane CENTRAL (Cochrane Library), Embase (OVID), Medline (Elsevier), PsycInfo (OVID), Social Policy and Practice (OVID), and Science Citation Index (Web of Science). EPPI-Reviewer 4, Endnote and Microsoft Excel were used to sift citations. Search terms included relevant Medical Subject Heading (MESH) terms and text words for sexual minority identity, behaviour and relationship status. 

In addition to database searches, reviews and summaries of lesbian, gay, bisexual and trans (LGB&T) health were examined for additional evidence to ensure all relevant studies were included. Hand search of several relevant journals was conducted (Journal of Homosexuality (2017–June 2018), LGBT Health (2017–June 2018) Journal of LGBT Health Research (all issues), Journal of Lesbian Studies (2014–2018) and Journal of Gay and Lesbian Mental Health (2014–2018)) as different journals are indexed in different databases and entry time varies.

Previous projects by the first author (CM) were sifted for relevant research and, from a previous project, a list of active LGBT health researchers and their publications were reviewed. Web pages of several researchers and organisations who had published health research in SMW were searched. The UK National LGB&T Partnership monthly newsletter from February to August 2018 was sifted to find recent unpublished research. UK national survey websites were examined for relevant information on SMW health (for example, Health Survey for England, Integrated Household Survey, Scottish Health Survey, Welsh Health Survey). 

### 2.2. Study Selection, Data Extraction, Quality Assessment

Full text copies of studies that may match the inclusion criteria were obtained. Two reviewers (CM and RH) checked study eligibility. For quantitative data one reviewer independently extracted data from studies into tables (CM) and these were checked by another reviewer (AM), with disagreements resolved through discussion. For qualitative studies relevant results were copied from the included studies into a separate document for reorganisation by descriptive themes. Characteristics and results of included studies were described. (See Table 1 for characteristics of included studies and Table 2 for quantitative results). The Critical Appraisal Skills Programme (CASP) qualitative studies checklist was used to assess quality of interview and focus group studies (Table 3). The question on the CASP qualitative checklist not having yes/cannot tell/no responses was omitted (i.e., question 10 on the value of the research). The CASP checklist for cohort studies was used to assess quality of the quantitative studies in order to give consistency in quality assessment strategy across studies (see Table 4). Questions on this checklist not having yes/cannot tell/no responses were omitted (i.e., study results and their precision, and implications of the results) as these are reported in the results section where appropriate. Studies providing both qualitative and quantitative results were assessed with both checklists. The Confidence in the Evidence from Reviews of Qualitative Research (CERQual) approach [17] was used to summarise our confidence in the systematic review findings across the included studies (Table 5). The review finding headings in the text of the results section correspond to the CERQual assessments in Table 5. 

### 2.3. Synthesis Methods

Synthesis of the quantitative results was through narrative description and tabulation. Meta-analysis was not appropriate due to heterogeneity of study designs and outcomes measured. Synthesis of qualitative studies was through thematic synthesis. One researcher (CM) extracted all quotes and author’s analyses from the included studies, coded them and organised them into descriptive themes. A second researcher (RH) independently coded the quotes and author’s analyses and organised them into another set of descriptive themes. Both researchers together then used the two sets of descriptive themes they had developed to establish analytical themes. These were then reanalysed by the second researcher, who selected illustrative quotations from the original studies to be reported alongside analytical themes. CERQual analysis was then used to develop the finally reported themes. Both researchers had experience analysing qualitative research, one through conducting systematic reviews (CM) and one from conducting primary qualitative research (RH). Neither (CM) nor (RH) had been involved in the conduct of any of the included studies. Combining the qualitative and quantitative results was undertaken in the discussion section, in order to give meaning to the body of evidence as a whole. 

## 3. Results 

A total of 23,103 citations were identified, 22,763 from the first searches and 340 from the second searches (see Figure 1). Full texts of 692 papers were screened for potential relevancy. There were 26 studies included, described in 29 papers, of which 22 provided qualitative results and nine provided quantitative results (studies providing both quantitative and qualitative results were [18,19,20,21,22]. The main reasons for exclusion were that results were not given separately for women and that the papers were not on experiences of UK healthcare. For a full list of references to included studies please see Appendix A.

Characteristics of included studies are described in Table 1. Participants in the studies were from the general community and varied in ages from schoolchildren [23] to over 50 [22]. Some of the studies were very large [24] and some compared results from lesbians and bisexual women or SMW to heterosexual women [25] whereas others were small and some recruited lesbians only [26]. The service areas varied from describing experiences of general health services [19] to describing very specific services such as cancer care [27], sexual health services [28] or midwifery [29]. Nine studies provided quantitative results (Table 2) of which two also contributed qualitative results [20,22]. In total, 22 studies provided qualitative results. 

### 3.1. Qualitative Study Results 

#### 3.1.1. Unhelpful Health Ambience

One theme which emerges strongly from the literature regardless of the types of health care provided is the physical context and ambience of the interaction. The patient journey was fraught with expectations of heteronormativity (assumption of heterosexuality) throughout, but initial impressions given by the images in waiting areas, leaflets, forms to be completed, and vocabulary used by staff members were likely to set the tone for any consultation. The visual and non-verbal environment created as a patient progresses through the system can be supportive and enabling, or it can reinforce that their identity is not recognised and give a perception of exclusion. Simple changes to promote visible inclusion of SMW makes a huge difference however the current reality was overwhelmingly that images, leaflets and language were identified by women as making assumptions of heterosexuality.

With respect to forms, for example, one patient felt that her legal relationship was devalued four years after the advent of civil partnership:

“The booking clerk asked me about my marital status. I said I’m civil partnered, she said what’s that? I said this is my partner we are in a civil partnership. She said I’ll put you down as single” [27] (p. 6).

Leaflets available in waiting areas, pictures on walls and information leaflets equally failed to depict diversity:

“They were all very heterosexual and there was absolutely no mention of a gay relationship or partners. So it didn’t feel it was; it didn’t feel it could be about me” [30] (p. 298).

Alternatively, leaflets were simply inappropriate; women reported having to ’translate’ information to make it appropriate to their situations.

“We were given a print out of a document that would help a straight couple having problems having children, information included for example that ‘you should be having sex regularly’. This clearly does not relate to our situation at all” [19] (p. 15).

Respondents to Fish [26] concerning cancer care similarly found the ambience in waiting rooms and support groups to be alien, and focused on aspects of life which they felt were not relevant to them.

For women seeking acceptance, appropriate leaflets and posters with the inclusion of diverse imagery and content would be signifiers that a service was LGB(T) friendly and safe, and contribute to a positive consultation experience. Several different lesbian respondents in River [22] commented on the desirability of indicating the service’s openness by visual means such as posters depicting same sex couples, and commented that LGB specific leaflets would provide useful information for women who were not part of the LGB community and who had little other access to LGB specific health information. As Westwood [31] points out, heterosexuality is privileged by the absence of images and leaflets which include LGBT people. Findings such as these were confirmed by other researchers for example Carter et al. [32] in the context of maternity services and Cherguit et al. [29] in the context of co-mothering.

One respondent suggested that LGBT specific leaflets were actively removed from waiting areas. A lesbian respondent [22] saw the sudden disappearance of Broken Rainbow (domestic violence in same sex relationships support service) from the General Practitioner (GP) surgery as possible evidence that LGBT specific leaflets were thrown away or hidden.

Respondents were not entirely negative; many who were accessing fertility clinics praised the LGBT friendliness and one women particularly wanted to be a participant in Cherguit et al.’s [29] study in order to record her positive experience throughout the process.

Ambience is important as it sets the tone for the rest of the interaction with the service and impacts on what follows in terms of women’s expectations of welcome or prejudice. 

#### 3.1.2. Assumed Heterosexuality/Heteronormativity

It could reasonably be assumed that a lack of LGBT friendly images and leaflets meant that staff did not have lesbian and bisexual women in mind when providing a service and this inevitably led to heterosexist assumptions in personal interactions. Respondents reported that language used by staff during consultations was experienced as exclusive [20,27,29], and required women to contradict assumptions in order to come out, creating a power dynamic, which some women reported as disabling [19,33]. Some women commented that it could be difficult to identify whether they were experiencing overt discrimination due to their sexual orientation, or simply poor practice which would have been similar, although differently expressed, regardless of their sexuality [29,33] 

Assumptions of heterosexuality were likely to be influential in different ways. Firstly, women felt unwelcome and that the service, whatever it was, was not aimed at them [27,33,34]; in many instances this would then influence women’s decisions as to whether or not to be open about their sexuality [20,33,35]. Secondly assumptions were made about what it was to be a lesbian or a woman who has sex with women [33]. As a result of these assumptions, relationships with professionals were considered to be less good than they might have been, women felt less able to discuss their sexual orientation and therefore the clinicians were unable to make holistic decisions about care and support; This in turn could have resulted in less good (medical) care being provided [19,20,27,32].

Basic expressions of heterosexism (overt or covert discrimination on the grounds of not being heterosexual) were reported by women in many studies. Typically, this included failure by staff to recognise the same sex partner as that, a partner. 

“On the day, the locum firstly ignored my introduction as ‘partner’ and continued to call me ‘friend’ for the rest of the session” [19] (p. 16).

Even when the evidence of the partner was physically present, professionals apparently found it difficult to treat or speak to female partners in the same way as they would have treated or spoken to husbands or male partners. Again this is evidenced across many services, such as ante-natal classes:

“Kept saying ’right, mums over here, dads, I mean or partners’, so she said ’dads, I mean partners!’ about 74 times before she finally got her head around just saying partners” [29] (p. 1273).

Issues around the inclusion of same-sex partners in consultations were often mentioned regardless of the setting. A number of participants described instances where partners were negated or derogated [33]. The acceptance of same sex partners was particularly important as women wanted their partners recorded as next of kin and to be the person making decisions for them if required [27,33]. Many felt that their relationships were not recognised:

“(The receptionist) refused to put down my partner’s name and partner/next of kin, kept saying ‘I’ll just put friend’, I said, no, I want you to put partner and she looked at me all lips pursed and said, ‘I’ll just put friend.’” [19] (p. 16).

Heterosexist norms and systems were often applied routinely without adaption for non-heterosexual patients. For example, for lesbian patients, hair loss following chemotherapy meant that something as apparently simple as a wig fitting could become problematic. The only available hairstyles were long and very feminine and often inappropriate for some women’s usual style [33].

Although negative experiences outweighed the positive, neutral or positive interactions were reported where same sex partners were accepted without comment by all staff [19,29]. It is perhaps concerning that professionals not reacting negatively to a woman with a same sex partner was worthy of positive comment from respondents. Commenting on the services received for end of life care in a hospital setting, one woman reported:

“I actually found that all the agencies that I had to deal with were totally professional and really helpful and supportive.” [30] (p. 297).

#### 3.1.3. Being “Out” or Not

Whether or not women chose to be open about their sexuality with health professionals was a complex topic with many factors impacting on the decision. Coming out to professionals potentially impacted the physical and psychological treatment women received. How health professionals responded to women declaring their sexuality contributed to women’s overall experience of services received. In many cases this was influenced by the experiences of the service until the point of meeting the relevant health professional. Fear of prejudice or discrimination based either on previous experience or experience of friends meant that many women chose not to share information about their sexuality. Others chose to share information about their sexuality dependent on whether they thought that this would be medically relevant [20,32,33]. This could be problematic if women were seeking gynaecological treatment as they were unsure as to the relevance of their sexuality to the consultation [32]. If neither patient nor professional mention sexuality and so are unaware of possible health implications, then the potential for compounding the problem increases and the importance of this aspect of life in planning care is missed. 

Some women expressed a wish to maintain control of who knew about their sexuality and made a new decision about coming out with respect to each professional they met. Confidentiality was of particular concern when confidentiality policies were unclear [27,32]. Other women requested that their sexuality be recorded on their patient notes so that they did not continually have to come out, although this was not always possible, as in one instance a woman was told it was not information that was recorded in the personal details [32]. The power dynamic of “coming out” is clearly important to SMW and the persistent levels of sexual orientation hate crime and workplace discrimination remind us that disclosure is not without risk [36].

A common theme, regardless of the area of health care, was the awkwardness of coming out. Carter et al.’s [32] research into lesbian and bisexual women’s experiences of cervical screening comments that raising the topic could be difficult usually because of assumptions of heterosexuality, but other women in Humphreys and Worthington [19] identified lack of time in appointments as the influential factor. Additionally, women found that they were asked questions about contraception when the smear test was in progress, which was experienced as a particularly difficult time to discuss their sexual identity [20].

A common experience was that women felt that they were forced to be out; typically.

“I wouldn’t mind, but I didn’t really want to ‘come out’ to my nurse—she kept asking about contraception and sex—I had no choice but to tell her” [20] (p. 35).

For women who had not previously been open about a same sex relationship, there was the possibility of needing to change their usual practice at a time of ill-health or partner death and thus a time of vulnerability.

“The death of a partner becomes a very public thing so it’s an issue and it forces you into a situation you weren’t quite ready for” [30] (p. 295).

The result of not coming out might mean that women passively accepted the false assumptions being made about them; this could be uncomfortable, but for some women it provided a feeling of safety and was preferable as it avoided the potential for overt prejudice [32,35].

#### 3.1.4. Responses to Being Out

Although many women experienced neutral or positive responses from their healthcare professional, a worrying number received negative responses.

“One couple-counsellor from the agency claimed she could not understand me. She said that I was attractive, had everything going for me, and didn’t really understand what my problem was” [34] (p. 386).

Negative responses were frequently reported in the context of cervical screening:

“It was her face, I’ll never forget it but she was physically repulsed, and that is how it felt, she was absolutely appalled” [20] (p. 34).

Women also reported the professional gasping [22], physically recoiling [20] or receiving a lecture during an ultrasound of the necessity for a child to have both a mother and a father [19].

On the other hand, there were many reports across all settings of supportive practitioners; these were particularly prominent in the research about women’s experiences of being out in GP services [22].

Interestingly, women were occasionally uncertain whether a comment made was intended to be supportive or was homophobic. For example:

“When Jessica was born she said ‘oh aren’t you lucky you didn’t have a boy because you wouldn’t know how to deal with penis’ and it’s like ‘what!’ (laugh) you don’t expect that from a doctor” [29] (p. 1273).

The experiences of this consultation, previous health interactions and general experience of discrimination all contribute to the way in which ambiguous comments are understood.

#### 3.1.5. Ignorance

There were a worrying number of reports of medical ignorance with regards to SMW’s health. Many, but not all, of these examples were concerned with whether women should be undergoing cervical screening as health professionals did not agree amongst themselves about whether a smear test was required. This comment was typical of respondents’ experiences:

‘Nurse and doctor have always said I don’t need one—lesbians cannot get cervical cancer, so of course, I won’t go through an embarrassing procedure I don’t need’ [20] (p. 32).

Additionally, medical staff appeared ignorant about SMW’s sexual health in general. One woman who asked for dental dams rather than condoms was met with blankness, confusion and uncertainty [19]. On another occasion midwives seemed unable to differentiate between the two women in a couple, treating the one who was pregnant as if she had previously given birth when in fact it was her partner who had done so [26].

#### 3.1.6. Impact on Sexual Minority Women

The inevitable result of negative experiences was that women either delayed or did not access health care. Carter et al. [32] noted that some participants avoided healthcare of any kind whilst others had not registered with a GP or changed their contact details. “Two (women in this study) avoided going to the GP when they had a problem which resulted in delayed treatment” [32] (p. 137).

For others the treatment might have been less good, for example following a mastectomy:

“The decision not to have reconstruction meant that the consultant did not perform the operation and this led to a reduction in the quality of her surgery” [27] (p. 15).

Likewise, in a counselling context, lack of knowledge was perceived to impact negatively [34].

In consultations, an atmosphere of discomfort and embarrassment, regardless of the vocabulary used could result in patients and partners feeling unable to take full advantage of the consultation and thus received a potentially less good service:

“If we’d had someone treating us that was maybe, was very relaxed about, you know, our sexuality, or whatever, I think it might have just made it a bit easier to ask questions” [27] (p. 5).

Finally, negative experiences added to feelings of being marginalised or different with the potential for associated loss of confidence and self-esteem:

“If you were feeling bad about yourself, you’ve got low self-esteem or, you know, had the experience of homophobic abuse, and then you went somewhere and you couldn’t find the information you wanted, it kind of reinforces the difference” [27] (p. 17).

Affirming responses result in better consultations. Two women in the Humphreys and Worthington [18] study reported that they would ask more questions on the next visit, or feel confident to see the professional again with any future issues.

#### 3.1.7. Challenging/Complaining

When experiencing what they considered to be discriminatory language or treatment, women considered complaining but rarely did so. One woman highlighted a variety of reasons for not complaining:

“There’s also that thing of if you complain do you, you know, you get branded in some way (laugh) and it was, also its also a structural thing, so its not that anyone, you know, I couldn’t say that person was homophobic and complain about them” [29] (p. 1273).

Another had no confidence that a complaint would be taken seriously and raised an important point:

“Looking back I should have complained about her, but didn’t feel confident enough—what if the person I complained to was just as homophobic” [20] (p. 33).

Willis et al. [37] in their research with LGBT carers note that “Overt experiences of discrimination were considered not worth reporting because of the emotional resources required to challenge discriminatory treatment from health care professions” [37] (p. 1312). Complaining was often seen as an unavailable option as it might lead to less favourable treatment. 

#### 3.1.8. Bisexual and Trans Participants

Of the 22 studies included in this review, bisexual respondents were included in 19 studies and women who identified as trans were included in six studies. We have chosen to report bisexual and trans women’s experiences separately to ensure that their specific experiences are represented. Most of the issues raised by bisexual and trans women were similar to issues raised by women who identified as gay, lesbian or queer, for example complaints about insensitivity including assumptions about the implications of their self-definition as bisexual or trans, but the issues impacted on them differently.

Some women pointed out that their bisexuality was invisible; women were sometimes disbelieved. One woman currently in a relationship with a woman was assumed to be a lesbian despite her otherwise respectful treatment and her insistence that she was bisexual [19]. Another woman describes feeling hurt when asked if she had ’switched sides’ [19] and a woman accessing counselling felt that the counsellor actively denied her bisexuality and wanted her to realise that she was really straight [35]. If a woman had a woman partner at the time of the consultation, it was assumed that she was a lesbian and did not/had not had sex with men, an assumption that could be medically risky and denies the validity of bisexual identity.

There is very little research or acknowledgement of trans SMW and what limited research has been undertaken into trans women’s experiences focuses on their gender identity rather than their sexual orientation. A vital issue for lesbian or bisexual transwomen was their gender status. For those who were ill or coming towards the end of their life, the urgency for being treated and dying as women was crucial:

“I’m not ready to die. I want my surgery first, and I was hanging on in there. It was important to me to be buried as a woman, not half and half, you know, with the physical side of it” [38] (p. 27).

Young people reported long waits for appointments at gender assignment clinics which impacted on their mental health:

“Yeah, it took a month… it took a month for the… for the referral to sort of like be processed by them and then their response was, ‘We can’t see you for six months,’ which obviously, you know, started making me feel about the same again from before [suicidal]” [39] (p. 65).

Lack of respect for women’s status took many forms, including failure to use the correct pronoun [38], this was sometimes then extended as clinicians struggled to process non-heterosexual identities of trans women. There were frequent reports of gender not being recognised:

“In 2008 I had knee surgery and woke up on a male ward—clearly they had looked at my face and overruled my notes” [19] (p. 16).

Women in this group were also questioned and treated inappropriately:

“Bearing in mind I had given him my history, he actually asked me about my periods” [19] (p. 16).

And on another occasion: 

“I was scheduled for a small bit of surgery and was asked to give a pregnancy test. I pointed out that I was not only a gay woman but also post-op male-to-female trans. The reply was ‘Well, best to be sure’” [19] (p. 16).

Lack of awareness resulted in ‘outing’ women:

“I’ve been in resus where I didn’t know if I was going to survive or not... just with curtains. And you can hear every conversation...Some doctors have said to me, ‘How long have you been transgendered for?’ And everybody has heard” [38] (p. 29).

A lack of realisation that following usual protocols would impact disproportionately on trans women was reported. In one instance, detained in a psychiatric hospital, in addition to taking no action to make her feel comfortable as a trans person, a woman was not allowed a razor, so her beard grew, to the inevitable detriment of her mental health [37]. 

### 3.2. Quantitative Comparative Results

Four included studies compared results for SMW and heterosexual women [18,23,25,40]. They tended to show SMW had worse experiences when accessing healthcare (see Table 2). For example Elliott et al. [25] published an evaluation of the English General Practice Patient Survey by gender and sexual orientation. The weighted percentages reporting no trust or confidence in the doctor was 5.3% (95% CI 4.7 to 5.9) in lesbians and 5.3% (95% CI 4.6 to 6.0) in bisexual women, compared to 3.9% (95% CI 3.8 to 3.9) in heterosexual women. Both differences were statistically significantly worse for SMW. There was also significantly worse doctor communication and nurse communication. More SMW were fairly or very dissatisfied with care than heterosexual women and for lesbians this was statistically significant. 

Urwin and Whittaker [40] published another evaluation of the English General Practice Patient Survey, looking at inequalities of GP use by sexual orientation. They found that lesbians and bisexual women were less likely to visit the GP than heterosexual women in the previous 3 months (adjusted OR = 0.80 (95% CI 0.76 to 0.85 and OR = 0.89 (95% CI 0.82 to 0.96)) and this was not affected by the proportion of GPs who were women. On the other hand, a survey of schoolchildren in Cambridgeshire [23] found that 84% of sexual minority girls had been to the doctor’s surgery in the previous 6 months compared to 76% of Cambridgeshire girls, and that 34% of sexual minority girls had felt uncomfortable or very uncomfortable talking to the doctor or other surgery staff compared to 26% of Cambridgeshire girls.

### 3.3. General Experience of Health from Non-Comparative Studies

A very large survey of LGBT experiences of everyday life in the UK [24] included 108,100 responses (see Table 2). Most of the chapter on health gave numerical results for men and women combined, but there were some results for SMW, but only for cisgender rather than both cisgender and trans women. The results showed widespread difficulties with accessing services, including for mental health and sexual health. 

A survey commissioned by the LGBT Partnership [19] on SMW experiences of healthcare found that the majority were of GP/primary care (51%) but also included hospital (33%), sexual health clinic (14%), mental health (6%), fertility clinic (2%) and dentistry (1%). There were more negative experiences in mental health services and hospitals than sexual health clinics and GP/primary care services. The majority of the negative experiences reported took place in the previous year to the survey (i.e., 2014–2015). The main themes for the negative experiences were assumption of heterosexuality, clinicians being uncomfortable with minority sexual orientation, participants being given incorrect or incomplete information based on sexual orientation, bad treatment (possibly) not related to coming out, partner not being acknowledged, experience of overt homophobia or biphobia, or clinicians ignoring the patient disclosing their sexual orientation.

A survey of older LGBT people [22] found that 43% of SMW had had good experiences with their general practice and 31% reported bad experiences. These included overhearing homophobic comments, overt prejudice from a GP towards their partner with cancer, assumptions of heterosexuality by all staff including receptionists, lack of awareness of SMW’s issues, partners being ignored, shock and embarrassment by health staff on disclosure, and inappropriate disclosure of sexual orientation to a third party.

Two UK studies were found on cervical screening attitudes and uptake in lesbians and bisexual women [20,32] and one provided quantitative results. Light et al. [20] conducted a multi-method evaluation of a project delivered by the then Lesbian and Gay Foundation (LGF - now LGBT Foundation) with SMW in the Northwest of England. From the survey, although 91% agreed that SMW should have cervical screening, only 70.5% of those eligible had accessed screening in the previous five years and 48% within the previous 3 years. There was clear evidence found that SMW had been misinformed by being told they did not need a cervical smear and 14% of those eligible had been actively refused or discouraged from having a smear test by a health professional as a direct result of their sexual orientation. When they did attend, many SMW were subjected to heteronormative assumptions. Following this survey, a public information campaign was run by LGF called ‘Are you ready for your screen test?’. This was well received and accepted by lesbians and bisexual women in the North West and was evaluated by a second survey with 345 responses. The campaign resulted in an additional 22% of those aged 25 or more having gone for a cervical screen and a further 8% having booked a cervical smear appointment. 

## 4. Discussion 

### 4.1. Summary of Findings

This rigorously conducted and innovative mixed-methods systematic review included 26 studies, of which 22 provided qualitative results and nine provided quantitative results (two studies provided both quantitative and qualitative results [18,19,20,21,22]). All included studies were relevant to the delivery of UK healthcare services. A major strength of the findings is the demonstration of consistency across studies, including studies generated by small organisations and by the UK government, and the coherence of findings across qualitative and quantitative studies. This is systematic review is innovative in that there are very few mixed-methods systematic reviews and there have been no previous systematic reviews of SMW’s experiences of UK healthcare. It is also one of very few to incorporate CERQual assessment of outcomes (Table 5). 

In addition to the protections afforded by the Equality Act (2010), the National Health Service (NHS) constitution states that “Respect, dignity, compassion and care should be at the core of how patients... are treated”. Although some women in specific services reported that this was the case, the majority of women included in these studies reported otherwise. Years of experience of prejudice means that women need positive signs/images that a service will be LGBT friendly. Negative expectations were confirmed by a plethora of experiences such as the ambience of the service and the attitude of reception staff, inappropriate protocols that needed to be followed, language used, assumptions made and apparent ignorance of SMW’s health needs. 

First impressions are important, thus images and leaflets in waiting areas set the tone of what could be expected. What might appear as a minor issue to others has a greater impact on those who have experiences of discrimination—images and the use of language are important in building up a trusting atmosphere. In many instances it is this pervasive heteronormativity that directly influences women’s decisions on coming out or not to the professional they see, and therefore potentially limits their ability to receive holistic care. Systems that allowed appropriate registration of same-sex partners, and attitudes of reception staff prior to a consultation with the relevant professional, all contributed to women’s assessment of whether they would be treated respectfully and their identities meaningfully recognised. Appropriate posters and leaflets are important, but if a service provides these, expectations are raised and the agencies would then need to ensure that a service that fulfils those expectations is provided. Inclusion and openness which is tokenistic is likely to have a detrimental outcome.

Health professionals were seemingly unable to adapt information given and procedures followed to SMW’s specific situations; this was particularly obvious in fertility clinics and cervical screening. However, at times of reconstructing their self-image, for example as a cancer patient, it is unhelpful if a woman’s physical style or style of dress and hair has to be amended to fit in with what NHS provision apparently considers to be the norm for women. Equally striking was the apparent ignorance of health professionals about SMW’s health needs as clearly demonstrated by the inconsistent information provided about cervical screening. Extremely worrying here is the increased medical risk to women as evidenced by confusing a woman who is a first time mother with her partner who has given birth, or ignoring her history and refusing a bisexual woman a smear test as her current partner is a woman.

It is essential to remember that interactions with services tend to occur at a time of difficulty, illness, vulnerability or crisis. Coupled with fear of discrimination, this is not a time when women are likely to feel able to challenge or complain about poor treatment, unthinking assumptions about them and their lives or apparent active homophobia. SMW reported heteronormative assumptions leaving them with the choice of either going along with these assumptions or challenging them and thus risking negative reactions and potential breaches of confidentiality. 

In order to form a trusting and open relationship with professionals, SMW need to feel respected for who they are. As is clear, the negation of partners, the use of inappropriate vocabulary and assumptions all militate against this. What SMW expect from the health professional is no different from what all patients expect and is promised in the NHS constitution. Health practitioners need to be aware that treating people equally and respectfully does not mean treating them the same, but making adjustments appropriate to their life situations. The assumed heterosexuality that SMW may encounter influences every aspect of their journey through services.

The impact of the experiences of marginalisation, labelling and direct discrimination cannot be underestimated. The way in which staff interacted with SMW might well be open to interpretation and many women expecting negative responses may thus interpret ambiguous responses negatively. It was also possible that the professionals in question were simply lacking in people skills so that all patients were treated equally poorly. It must be remembered that complaints voiced by many patients may impact differently on SMW; for example, meeting different doctors at every appointment in ongoing treatment means that women may constantly be deciding whether or not to come out, with the potential additional stress that this might entail. 

One further comment is on the use of the term ‘disclose’ when women are considering whether or not to share their identity with professionals. In current English usage this term carries negative connotations, for example in ‘disclosing’ a criminal record. Such language is unlikely to encourage women to be open about who they are.

The quantitative comparative studies demonstrated that SMW experience worse interaction with UK health and social care in a wide variety of settings and services than heterosexual women. The non-comparative studies, including one extremely large survey by the UK Government Equalities Office [24], found very worrying trends in difficulty with accessing a wide variety of health and care services.

### 4.2. Strengths and Weaknesses of the Systematic Review

A major strength of this systematic review is the combining of findings from qualitative and quantitative research. Other strengths include extensive searches from a number of different sources. We assessed quality of individual studies using CASP questionnaires appropriate to the different study designs, to give an element of consistency in questions about bias assessment across qualitative and quantitative studies. 

We used a wide definition of SMW including identity, behaviour and partnership. Although they are different concepts, (some women identify as lesbian whilst having sex with men, some women identify as heterosexual whilst having sex with women, and women can identify as lesbian or bisexual without being sexually active or being in a partnership) they are all representative of sexual minority status. The studies used self-report for the experience of healthcare and this may therefore result in responder bias, but it is unclear why responder bias might be stronger in SMW than heterosexual respondents. There is a potential conflict of interest where a charity or other small group seeks to demonstrate an issue in order to redress a wrong. 

Several studies combined results for men and women and thus picking out issues specifically related to women was challenging. In the qualitative systematic review we used direct quotations rather than narratives from the papers where the author’s analyses incorporated both men’s and women’s issues so that we could report the women’s experiences. We used rigorous methods to synthesise the findings from a large number of studies to generate themes applicable to multiple health care delivery situations. 

We also used CERQual [17] to generate evidence profiles of our findings to show how the relevance, coherence, adequacy and methodological limitations of individual studies impacted on our overall qualitative findings under each of the headings in the main text. 

### 4.3. Comparison to Previous Research

There have been previous systematic reviews on UK LGB health but none focusing on SMW and on experience of healthcare. There have been no previous mixed-methods systematic reviews in this area incorporating CERQual assessment of outcomes. A wide-ranging systematic review on health, education, employment, housing and other topics, [3] written for the UK Government Equalities Office, included small sections on the use and experience of mental health services, satisfaction with health care and discrimination, and recommendations for policy, but did not distinguish between men’s and women’s health experiences. An extensive overview of health needs of lesbian and bisexual women [41] looked at experiences and expectations regarding healthcare providers also found negative experiences, lower satisfaction and fewer than half of SMW being out to their GPs. SMW frequently reported that healthcare providers assumed they were heterosexual, and that they were not given a chance to ‘come out’. When women did come out this information was commonly ignored, and occasionally negative comments were made. 

There is a clear gap in research into bisexual women and trans SMW’s experiences, and this biases the perspectives to those of lesbian-identified women, especially in quantitative research where SMW are often combined for analysis due to limited sample size.

## 5. Implications and Recommendations for Practitioners

Many health care staff feel that they give person-centred care to all of their patients or clients including SMW, and therefore they do not need to know about their sexuality. A survey by the Stonewall Charity on the treatment of LGBT people within UK health and social care services [34] found a worrying amount of lack of knowledge and understanding of the issues, unfairness, negativity and some blatant discrimination by staff. 

There is a need to incorporate SMW issues into guidelines for healthcare. A systematic review of primary care guidelines for LGB people [42] included 11 guidelines (two from UK). They found that the currently available guidelines for LGB care are philosophically and practically consistent, and synthesised recommendations could be readily applied to existing primary care systems with minimal change and no cost to practice systems, but staff training would be needed. The Lesbian, Gay, Bisexual and Trans Public Health Outcomes Framework Companion Document [11] sets out the evidence base related to each public health indicator, and makes clear recommendations for action at local, regional and national levels. Regarding healthcare it recommends that 

“Commissioners should use the data available to them to assess whether mainstream services they have commissioned are accessible to and appropriate for LGBT people”. 

And also that

“Commissioners should ensure provision of specialist services, where appropriate, to address specific healthcare needs available in their local area.”

There is a need for including issues around care for SMW in medical, nursing and allied professional training curricula. A recent review of UK issues around nursing care [43] concluded that, although a number of studies internationally had investigated LGBT nursing care and how it could be introduced into the nursing curriculum, there were no recent UK studies. There was little attention paid to LGBT patients’ needs in many university nursing programmes, resulting in nurses being less than confident when nursing LGBT patients [43]. Concepts of homosexuality were difficult for nurses who were not being exposed to SMW, because SMW were not coming out in a nursing context. Experiences of lesbians should be made clear to staff to enable them to become familiar with the needs of this population and understand and modify the way they provide care. Health professionals also need to learn to abolish prejudice to enable them to deliver comprehensive and appropriate care.

## 6. Conclusions 

There is very little research published on SMW health [1] and even less on experiences of healthcare. This mirrors the general trend of little investment in LGBT research [12]. There is clear and consistent evidence, despite limited research, that SMW face barriers to accessing and experiencing positive care. There is a strong need to enhance healthcare professionals’ understanding of how to provide culturally competent care for LGBT people and to understand this group’s health needs. Despite the fact that the NHS has a sexual orientation information standard guideline and training to support implementation, changing attitudes is not straightforward. It is unclear how long it will take for equality and diversity messages to filter through to front line healthcare staff resulting in practice change. While the current status quo continues, SMW continue to receive poor and inappropriate care in many situations.

## Figures and Tables

**Figure 1 ijerph-16-03032-f001:**
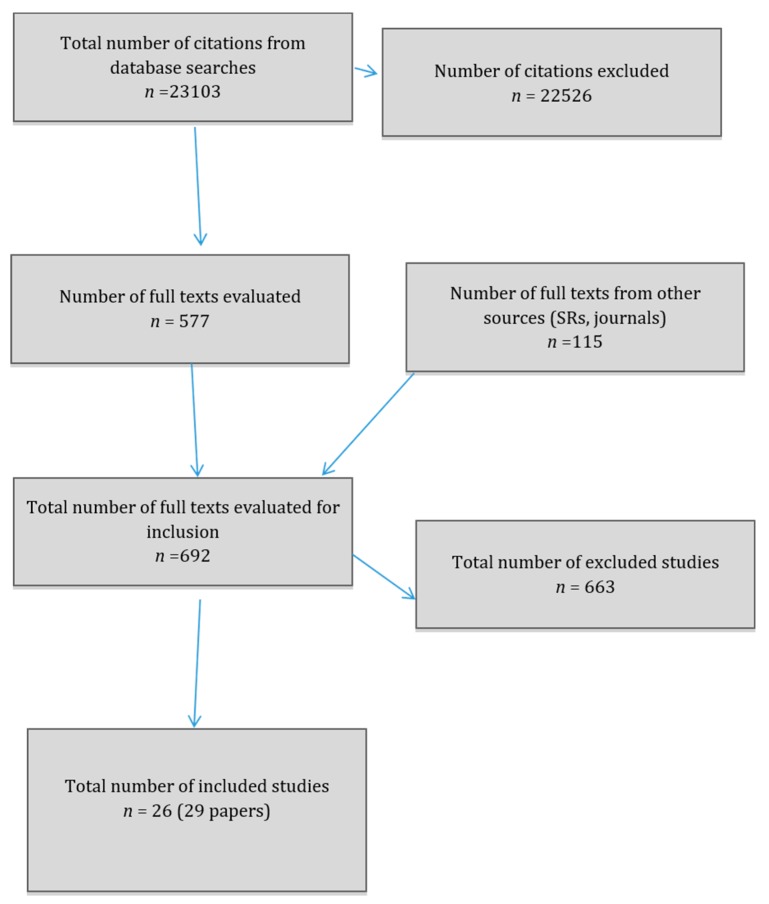
PRISMA* flow diagram. *Preferred Reporting Items for Systematic Reviews and Meta-Analyses.

**Table 1 ijerph-16-03032-t001:** Characteristics of included studies.

Study Author, Year	Study Design Method	Population, Setting	Number of Participants (Total Number in Study)	Recruitment	Sexual Orientation Ascertainment	Outcomes of Interest	Funding Publication Status
Almack et al. 2010	Four focus groups	LGB community	5 SMW (*n* = 15 total)	Unclear	Self-report	End of life care issues	Fully published, Funded by Burdett Trust for Nursing and Help the Aged (now Age UK)
Balding 2014	Health-Related Behaviour Survey	School year 10—aged 14–15	1916 Cambridgeshire girls, of which 92 LGBT (*n* = 3918 total)	Through schools	Self-report LGB	GP practice issues	Grey literature report. Schools Health Education Unit.
Bristowe et al. 2018	Semi-structured interviews	LGB community with advanced illnesses or their carers	18 SMW (*n* = 40 total)	Through palliative care teams (three hospital, three hospice), and nationally through social/print media and LGBT community networks	Self-report LGB	Experience of receiving care when facing advanced illness	Fully published. Marie Curie Research Grant Scheme
Carter et al. 2013	Individual and small group interviews	SMW in community	5 SMW (*n* = 5 total)	Unclear	Self-report	Cervical screening issues	Fully published, funding unclear
Cherguit et al. 2013	Semi-structured interviews	SMW in community	10 lesbian mothers (*n* = 10 total)	Via a donor conception charity then snowball.	Self-report lesbian	Midwifery and delivery issues	Fully published, not funded
Elliot et al. 2014	English General Practice Patient Survey 2009/10	Women in community attending GPs	1,021,541 women of which 0.6% lesbian, 0.5% bisexual. 86.1% heterosexual (*n* = 2,169,718 total)	Through GP surgeries	Self-report LGB using ONS categories	GP practice issues	Fully published, funded by UK Govt. Department of Health
Evans and Barker 2010	Survey (open-ended questions)	Community	47 women of which 44 SMW (*n* = 62 total)	Adverts including in Diva magazine	Self-report	Issues around mental health counselling	Fully published, funder unclear
Fenge 2014	Semi-structured interviews at home or workplace.	Community	1 lesbian (*n* = 4 total)	Snowball sample	Self-report	Bereavement experiences	Fully published, funding unclear
Fish 2010	Semi-structured interviews	SMW in the community with breast cancer or had partner with breast cancer	17 SMW (*n* = 17 total)	Flyers via networks, websites, email lists, LB women’s groups, cancer care services and Age Concern	Self-report	Breast cancer care experiences and issues	Grey literature, funded by National Cancer Action Team
Fish and Bewley 2010	Survey (open ended questions)	SMW in the community	5909 lesbian and bisexual women (*n* = 5909 total)	Promotional materials in gay and mainstream media and other distribution channels.	Self-report sexual minority	Nature of healthcare experiences, recommendations for improving services and any other healthcare experiences	Fully published, funded by Lloyds TSB Charitable Foundation
Fish and Williamson 2016	Semi-structured interviews	LGB people in the community diagnosed with cancer in previous 5 years	6 lesbians (*n* = 15 total)	Radio interviews, LGBT press articles, 50 local mainstream cancer groups, LGBT community-based groups, social media	Self-report LGB	Experiences of cancer care	Fully published, funded by Hope Against Cancer charity
Formby 2011 (and Formby 2011b)	Survey and focus groups	SMW in the community	54 SMW (*n* = 54 total)	Online and through local press, LGBT networks and commercial gay scene	Self-report	Sexual health services	Fully published, funder unclear
GEO 2018	Survey (online only)	LGBTI aged 16 or over	N women not given but approx. 45,402 (42%) (*n* = 108,100 total)	Via stakeholders, Pride events, national media, GEO, government social media, television interviews and online video	Self-report	Experiences of health services	Grey literature, funded by UK government
Guasp 2011	Survey	Older LGB and heterosexual, community	N women unclear, n SMW unclear. (*n* = 2086 total)	Through YouGov panel supplemented with social media campaign	Unclear	Future care (other results not presented by gender)	Grey literature report. Funded by Stonewall
Humphreys et al. 2016	Survey and 3 focus groups	SMW in the community	101 women (*n* = 101 total)	Through National LGB&T Partnership social media	Self-report	Healthcare experiences	Grey literature, funding unclear
Ingham et al. 2016	Semi-structured interviews	Older women in community	8 women who had lost a same-sex partner (*n* = 8 total)	Adverts to relevant charities, support groups and services	Self-report partnership status	Bereavement experiences	Fully published, funding unclear
Knocker 2012	Interviews	Older lesbians in community or sheltered housing	4 lesbians (*n* = 8 total)	Unclear	Self-report	Experiences of health and social care	Grey literature report, funded by Joseph Rowntree Foundation
Lee et al. 2011	Unstructured interviews	Lesbian mothers	8 lesbians (*n* = 8 total)	Snowballing from first participant	Self-report	Positive and negative experiences of maternity care	Fully published, not funded
Light and Ormandy 2011	Survey and 6 focus groups	Community	Survey 611 LGB women (*n* = 611 total), 60 in focus groups	Online survey, via Manchester Pride and Manchester Lesbian and Gay Foundation	Self-report	Cervical screening service experiences	Grey literature report, funded by NHS Cervical Screening Programme
Macredie 2010	Survey, with open and closed questions	LGBT in community	114 LB women (*n* = 212 total)	Convenience sample, including from pubs and clubs	Self-report lesbian/gay women or bisexual women	Fertility, screening (most results not split by gender)	Grey literature report. Commissioned by NHS Bradford and Airedale
McDermott et al. 2016	Survey and interviews	LGBT people in the community aged 16–25 years who had experienced self-harm or suicidal feelings, and mental health services staff	Survey 336 women (*n* = 789 total), interviews number of women unclear (*n* = 29 total)	LGBT organisations and social networks, LGBT mental health organisations	Self-report LGB or queer	Experiences of mental health services	Grey literature, funded by Department of Health Policy Research Programme
Price 2010 (and Price 2012)	Semi-structured interviews	LGB carers of people with dementia	11 SMW (*n* = 21 total)	Through Alzheimers’ Society then online fora, conference, advertising, word of mouth	Unclear	Experiences of dementia services	Fully published, funding unclear
River 2011	Survey (open and closed questions)	LGBT people aged over 50	144 SMW *n* = 283 total)	Through Polari Group mailing list, specialist websites, emails to community lists and social and campaigning groups in London	Self-report LGB	Experiences of GP services	Grey literature, funded by Age Concern England
Urwin and Whittaker 2016	English General Practice Patient Survey 20012/14	Women in community attending GPs	1,138,653 women of which 0.6% lesbian, 0.4% bisexual. 91.9% heterosexual (*n* = 2,807,320 total)	Through GP surgeries	Self-report LGB using ONS categories	GP practice use	Fully published, not funded
Westwood 2016 (and Westwood 2016b)	Semi-structured interviews	Older LGB in community or sheltered housing	36 SMW (*n* = 60 total)	Convenience sample via online adverts social networks, word of mouth,	Self-report various self-labels	Housing and residential care provision, concerns around dementia care	Fully published, funding unclear
Willis et al. 2011	Two focus groups and semi-structured interviews	Care stakeholders including carers	2 lesbian carers (*n* = 10 total)	Multiple channels including electronic fliers, Facebook, LGBT organisations	Self-report	Carers’ experiences	Fully published, University of Birmingham seedcorn funding

Abbreviations: GP—general practice; LB—lesbian and bisexual women; LGB—lesbian, gay and bisexual; LGBT—lesbian, gay, bisexual and transgender; ONS—Office for National Statistics.

**Table 2 ijerph-16-03032-t002:** Quantitative results.

Study		Lesbian	Bisexual	Mixed	Heterosexual/ Comparator	Statistical Significance	Notes
Balding 2014	Visited GP within previous 6 months	NG	NG	84% (77/92)	76% (146/1916)	NG	Comparator is Cambridgeshire girls
	Felt uncomfortable or very uncomfortable talking to doctor or other surgery staff on last visit	NG	NG	34% (31/92)	26% (50/1916)	NG	
Elliott et al. 2014	Trust and confidence in doctor = not at all	5.3% (95% CI 4.7–5.9)	5.3% (95% CI 4.6–6.0)	NG	3.9% (95% CI 3.8–3.9)	*p* < 0.001 both	Precise numbers for each question varied, numbers by sexual orientation not given. Adjusted percentages controlled for age, race/ethnicity, self-rated health, deprivation quintiles
	Doctor communication any item = poor or very poor	11.7% (95% CI 10.8–12.5)	12.8% (95% CI 11.9–13.7)	NG	9.3% (95% CI 9.2–9.4)	*p* < 0.001 both	
	Nurse communication any item = poor or very poor	7.8% (95% CI 7.1–8.4)	6.7% (95% CI 5.9–7.5)	NG	4.5% (95% CI 4.5–4.6)	*p* < 0.001 both	
	Overall satisfaction = fairly or very dissatisfied	4.9% (95% CI 4.3–5.5)	4.2% (95% CI 3.6–4.8)	NG	3.9% (95% CI 3.8–3.9)	*p* < 0.001 and p = 0.31	
GEO 2018	Did not discuss or disclose sexual orientation because afraid of a negative reaction	NG	NG	15.6% (cis)	NG	NG	Results given separately for cis and trans women. No heterosexual comparator for cis SMW. Nine percent of trans women were heterosexual, but results not given separately for SMW transwomen (or versus heterosexual transwomen)
	Did not discuss or disclose sexual orientation because had a bad experience in past	NG	NG	5.8% (cis)	NG	NG	
	Did not discuss or disclose sexual orientation because afraid of being outed	NG	NG	5.4% (cis)	NG	NG	
	Unsuccessful in accessing mental health services	NG	NG	9% (cis)	NG	NG	
	Rated access to mental health services ‘not at all easy’	NG	NG	27.4% (cis)	NG	NG	
	Experience of mental health services mainly or completely negative	NG	NG	22.2% (cis)	NG	NG	
	Accessing sexual health services not easy	31%	NG	NG	NG	NG	
	Had to wait too long to access sexual health services	NG	NG	12.1% (cis)	NG	NG	
	Was not able to go at a convenient time	NG	NG	11.5% (cis)	NG	NG	
	Worried, anxious or embarrassed about going to sexual health services	NG	NG	8.9% (cis)	NG	NG	
	Sexual health services were not close	NG	NG	7.1% (cis)	NG	NG	
	Did not know where to go to access sexual health services	NG	NG	5.9% (cis)	NG	NG	
	GP was not supportive	NG	NG	4.2% (cis)	NG	NG	
	GP did not know where to refer for sexual health services	NG	NG	2.3% (cis)	NG	NG	
	Experience of sexual health services mainly or completely negative	NG	NG	17.3% (cis)	NG	NG	
Guasp 2011	Experienced discrimination, hostility or poor treatment because of their sexual orientation when using GP services	NG	NG	17%	NG	NG	Numbers unclear, 40% of these incidents within previous 5 years
	Been excluded from a consultation or decision-making process with regard to their partner’s health or care needs	NG	NG	14%	6%	NG	Numbers unclear
	Hidden the existence of a partner when accessing services like health, housing and social care	NG	NG	12%	<1%	NG	Numbers unclear
Humphreys et al. 2016	Negative experience of GP/Primary care	NG	NG	47% (24/51)	NG	NG	Denominator numbers unclear
	Negative experience of hospital	NG	NG	66% (18/27)	NG	NG	
	Negative experience in a mental health setting	NG	NG	66% (4/6)	NG	NG	
	Negative experience in sexual health clinic	NG	NG	57% (8/14)	NG	NG	
Light and Ormandy 2011	Refused or discouraged from having a cervical screen by a health professional because of their sexual orientation	NG	NG	14% (70/500)	NG	NG	
Macredie 2010	Refused a cervical screen or advised it was not necessary	NG	NG	6% (7/114)	NG	NG	
	Found screening staff to be helpful but lacking in knowledge of lesbian and bisexual women	NG	NG	57% (33/62)	NG	NG	Of those screened
	Found screening staff to be unhelpful and lacking in knowledge of lesbian and bisexual women	NG	NG	12% (7/62)	NG	NG	
River 2011	Bad experiences of General Practice	NG	NG	31% (45/144)	NG	NG	
Urwin and Whittaker 2016	Odds ratio of visiting a family practitioner for any reason	0.803 (0.755–0.854)	0.887 (0.817–0.963)	NG	Referent	*p* < 0.001 and *p* = 0.004	Adjusted for patient and GP practice characteristics

Abbreviations: CEO—Government Equalities Office; 95% CI—95% confidence interval; cis—cisgender; GP—general practitioner; NG—not given.

**Table 3 ijerph-16-03032-t003:** Critical Appraisal Skills Programme (CASP) quality assessment of qualitative studies.

No	Study	1	2	3	4	5	6	7	8	9	10
1	Almack et al. 2010	Y	Y	Y	Y	Y	CT	CT	Y	Y	Y
2	Bristowe et al. 2018	Y	Y	Y	Y	Y	CT	CT	Y	Y	Y
3	Carter et al. 2013	Y	Y	Y	Y	Y	CT	CT	CT	Y	Y
4	Cherguit et al. 2012	Y	Y	Y	Y	Y	Y	Y	Y	Y	Y
5	Evans and Barker 2010	Y	Y	Y	Y	Y	Y	Y	Y	Y	Y
6	Fenge 2014	Y	Y	Y	Y	Y	N	Y	Y	Y	Y
7	Fish 2010	Y	Y	Y	Y	Y	CT	Y	N	Y	Y
8	Fish and Bewley 2010	Y	Y	Y	Y	Y	CT	Y	Y	Y	Y
9	Fish and Williamson 2016	Y	Y	Y	Y	Y	Y	Y	Y	Y	Y
10	Formby 2011	Y	Y	Y	Y	Y	N	Y	CT	Y	Y
11	Guasp 2011	Y	Y	Y	Y	Y	N	Y	CT	Y	Y
12	Humphreys et al. 2016	Y	Y	CT	Y	Y	CT	CT	N	Y	Y
13	Ingham et al. 2016	Y	Y	Y	Y	Y	Y	CT	Y	Y	Y
14	Knocker 2012	Y	Y	Y	CT	Y	N	CT	N	Y	Y
15	Lee et al. 2011	Y	Y	Y	Y	Y	Y	Y	Y	Y	Y
16	Light and Ormandy 2011	Y	Y	Y	Y	Y	CT	Y	Y	Y	Y
17	Macredie 2010	Y	Y	Y	CT	Y	CT	CT	N	Y	N
18	McDermott et al. 2016	Y	Y	Y	Y	Y	CT	Y	Y	Y	Y
19	Price 2015	Y	Y	Y	Y	Y	CT	Y	Y	Y	Y
20	River 2011	Y	Y	Y	Y	Y	N	Y	CT	Y	CT
21	Westwood 2016	Y	Y	Y	Y	Y	CT	Y	CT	Y	Y
22	Willis et al. 2011	Y	Y	Y	Y	Y	CT	CT	Y	Y	Y

Checklist questions were: 1. Was there a clear statement of the aims of the research? 2. Is a qualitative methodology appropriate? 3. Was the research design appropriate to address the aims of the research? 4. Was the recruitment strategy appropriate to the aims of the research? 5. Was the data collected in a way that addressed the research issue? 6. Has the relationship between researcher and participants been adequately considered? 7. Have ethical issues been taken into consideration? 8. Was the data analysis sufficiently rigorous? 9. Is there a clear statement of findings? 10. How valuable is the research? Abbreviations: Y—yes; CT—cannot tell; N—no; N/A—not applicable.

**Table 4 ijerph-16-03032-t004:** CASP quality assessment of quantitative studies.

	Study	1	2	3	4	5a	5b	6a	6b	9	10	11
1	Balding 2014	y	y	y	ct	ct	ct	n/a	n/a	y	y	n/a
2	Elliott et al. 2014	y	y	y	y	y	y	n/a	n/a	y	y	y
3	GEO 2018	y	y	y	y	y	ct	n/a	n/a	y	y	y
4	Guasp 2011	y	y	y	y	ct	n	n/a	n/a	y	y	y
5	Humphreys et al. 2016	y	ct	ct	y	ct	n	n/a	n/a	n	ct	y
6	Light and Ormandy 2011	y	y	ct	y	ct	n	n/a	n/a	y	y	y
7	Macredie 2010	y	ct	ct	y	ct	n	n/a	n/a	y	y	y
8	River 2011	y	y	ct	y	ct	n	n/a	n/a	y	y	y
9	Urwin and Whittaker 2016	y	y	y	y	y	y	n/a	n/a	y	y	y

Checklist questions were: 1. Did the study address a clearly focused issue? 2. Was the cohort recruited in an acceptable way? 3. Was the exposure (SMW status) accurately measured to minimise bias? 4. Was the outcome accurately measured to minimise bias? 5a. Have the authors identified all important confounding factors? 5b) Have they taken account of the confounding factors in the design and/or analysis? 6a. Was the follow up of subjects complete enough? 6b. Was the follow up of subjects long enough? 9. Do you believe the results? 10. Can the results be applied to the local population? 11. Do the results of this study fit with other available evidence? Abbreviations: y—yes; ct—cannot tell; n—no; n/a—not applicable.

**Table 5 ijerph-16-03032-t005:** CERQual qualitative evidence profile.

	Summary of Review Findings	Qualitative Studies Contributing*	Methodological Limitations	Relevance	Coherence	Adequacy	Assessment of Confidence in the Evidence	Explanation of CERQual Assessment
1	Unhelpful health ambience.Women reported that the environment did not include them	3,5–7, 10,12,14,17,19,20	Minor methodological concerns due to sample size of some studies and some data coding and analysis undertaken by only one researcher	Very minor concerns. Some studies are extremely local, but the studies together present a coherent picture	Very minor concerns as data consistent within and across studies	Very minor concerns despite low number of participants in some studies. Studies together provide rich data	High	This finding was graded as high as together these 10 studies present a coherent picture of women’s experience. Larger studies confirm findings of smaller studies. Rich data supports findings.
2	Assumed Heterosexuality /Heteronormativity	2,4– 12,17,20	Very minor methodological considerations due to lack of clarity concerning researcher role and potential bias in design and analysis of most studies.	Very minor concerns. Some studies very local or in big cities,	Very minor concerns. Findings are consistent within and across studies	Minor concerns due to small sample size of some studies. Larger studies provide very rich data and confirm findings of smaller studies.	High	This finding was graded as high despite very minor concerns in a minority of studies as together these studies provide rich data from a wide variety of settings. The 12 studies included provide a consistent picture regardless of service setting and service user group
3	Being Out or not	1,3–14,16,19,20	Very minor methodological considerations due to lack of clarity concerning researcher role and potential bias in design and analysis of most studies.	Very minor concerns. All demonstrate relevance to overall topic	Very minor concerns. Consistency across studies demonstrated. Data support findings	Studies together provide rich data across a variety of health and social care settings	High	This finding was graded as high despite some studies having a small number of participants as there was consistency of findings regardless of setting, geographical location and service user group. Sixteen studies contributed to this finding and rich data were evidenced
4	Responses to Being Out	4,5,7–9,12,15–17,22	Very minor methodological considerations due to lack of clarity concerning researcher role and potential bias in design and analysis of most studies.	Very minor concerns. All demonstrate relevance to overall topic	Very minor concerns. Data consistent within and across studies	Minor concerns due to sample size in some studies which offered little data about women’s experience,	High	This finding was graded as high despite minor concerns as ten studies contributed to this theme and larger studies provided consistent, rich data which supported the findings of smaller studies
5	Ignorance	3,5,8,10,12,15–17,22	Very minor methodological considerations due to lack of clarity concerning researcher role and potential bias in design and analysis of most studies.	Very minor concerns. All demonstrate relevance to overall topic	Very minor concerns. Data consistent within and across studies	Minor concerns as some studies were aiming to improve particular services.	High	This finding was graded as high as nine studies contributing to this theme provided rich data to support findings. Consistency and relevance across the studies assures the findings.
6	Impact on SMW	2,3,7,10–13,15,16,20	Very minor methodological considerations due to lack of clarity concerning researcher role and potential bias in design and analysis of most studies.	Minor concerns. All demonstrate relevance to overall topic	Very minor concerns. Data consistent within and across studies	Moderate concerns as half of these studies were categorised as ‘grey’ literature and half had small numbers of participants	Moderate-High	This finding was graded as moderate to high as a half of the studies were categorised as grey literature and half had relatively small numbers of participants. Despite this, data were consistent across studies.
7.	Challenging/ Complaining	4,7–9,12,16,20	Very minor methodological considerations due to lack of clarity concerning researcher role and potential bias in design and analysis of most studies.	Minor concerns	Very minor concerns. Data consistent within and across studies	Minor concerns, as this theme was not the focus of studies in most cases and the data were moderately rich	Moderate	This finding was graded as moderate. The data were consistent but lacked richness.

* Numbers here refer to the studies in Table 3 – CASP assessment of qualitative studies, rather than the reference list.

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
