# Peer review of "A Systematic Review of Sexual Minority Women’s Experiences of Health Care in the UK"

_ijerph, 2019, doi:10.3390/ijerph16173032_

Round 1
Reviewer 1 Report
the article is very valuable and is well written and researched.
I would suggest to the authors that more information on homophobia and heteronormativity needs providing at the outset. The readers of this article may not be well acquainted with these concepts and a clearer explanation (with relevant further reading/references) needs providing at the outset particularly for the benefit of medics who are not generally well informed about socio-cultural concepts.
Otherwise the article is publishable with this minor revision.
Author Response
Reviewer 1:
The article is very valuable and is well written and researched.
I would suggest to the authors that more information on homophobia and heteronormativity needs providing at the outset. The readers of this article may not be well acquainted with these concepts and a clearer explanation (with relevant further reading/references) needs providing at the outset particularly for the benefit of medics who are not generally well informed about socio-cultural concepts.
Otherwise the article is publishable with this minor revision.
Response:
We thank the peer reviewer for their positive comments. We are reluctant to reference the concepts of heteronormativity and homophobia in the journal article when they are well recognised as terms and defined in Wikipedia (see https://en.wikipedia.org/wiki/Heteronormativity , https://en.wikipedia.org/wiki/Homophobia ) We have given examples in medical scenarios which should help to illuminate.
Reviewer 2 Report
This mixed methods systematic review aims to evaluate the currently existing literature regarding health experiences of sexual minority women in the United Kingdom. This study is novel in that it presents data specific to this population in the UK. The study was conducted by researchers in the UK, and consists of a literature review and analysis of works, both published and unpublished, obtained from a variety of sources, including international databases, reviews and summaries, previous projects by the study's first author, works and web pages, from known researchers in the field, and several UK-specific LGBT websites. The authors evaluated both quantitative and qualitative studies containing data regarding experience of sexual minority women in the UK, and present their findings in this study.
Major comments:
There are several areas in the manuscript where statements are not backed up by citations. For example, lines 187-190 make an assumption regarding the reason women choose either to come out or not. Lines 192-199 similarly assume and suppose, rather than presenting evidence from the literature to back up the statements. Lines 200-207, 232-234, 297-298, 320-322, and 461-462 also do this. All these sections should have evidence to back them up, or perhaps be moved to the discussion section.Minor comments:
Line 64: the statement regarding the value of mixed methods compared to other types of studies should be backed up by a reference, unless this is an opinion of the authors, in which case this should be stated. Line 73: in reference to unpublished research, the authors may wish to state specifically where they obtained this unpublished research, and to which studies they are referring. Section 2.1: the authors very completely state where they obtained the studies included in their manuscript. A chart or figure showing how many studies were obtained from each source evaluated would add to this section of the paper. 2.3: I appreciate the explanation for why meta-analysis was not the chosen form of analysis. Lines 128 and 139 state that 22 and 23 qualitative studies, respectively were analyzed. Elsewhere in the paper 22 studies are sited; correction of this discrepancy is needed. Line 239 uses two separate tenses: chose and is. Consolidating the tense may be useful for readers to more clearly read this section. 3.1.8: the reasoning given for why bisexual and trans women were analyzed separately was useful; I appreciate the clarification. Line 529: the strength mentioned is a strength of the data, rather than a strength of the review itself. This should be mentioned. Rather than having two separate strengths and weaknesses sections, it would make the manuscript more streamlined to combine these into one section.Author Response
Reviewer 2:
This mixed methods systematic review aims to evaluate the currently existing literature regarding health experiences of sexual minority women in the United Kingdom. This study is novel in that it presents data specific to this population in the UK. The study was conducted by researchers in the UK, and consists of a literature review and analysis of works, both published and unpublished, obtained from a variety of sources, including international databases, reviews and summaries, previous projects by the study's first author, works and web pages, from known researchers in the field, and several UK-specific LGBT websites. The authors evaluated both quantitative and qualitative studies containing data regarding experience of sexual minority women in the UK, and present their findings in this study.
Major comments:
There are several areas in the manuscript where statements are not backed up by citations. For example, lines 187-190 make an assumption regarding the reason women choose either to come out or not. Lines 192-199 similarly assume and suppose, rather than presenting evidence from the literature to back up the statements. Lines 200-207, 232-234, 297-298, 320-322, and 461-462 also do this. All these sections should have evidence to back them up, or perhaps be moved to the discussion section.
Response:
Thank you for making us look again at some of these statements. For the most part we feel that they are appropriate for the results section as they describe the quotes coming after, summarise some of the findings or summarise that section. We had to be selective in the quotes that we used as the article was already becoming rather long. We explain below about each of the highlighted sections.
In original version: lines 187-190
We feel that the sentence about ambience is an appropriate concluding sentence for that section. We have moved the sentence about heteronormativity to the discussion section (new version lines 478-80)
In original version: lines 192-199 and 200-207
Although we didn’t use quotes to illustrate these points these issues were reported by a number of the studies and do represent some of the findings. We have edited the paragraph and added references to strengthen these links to the texts.
In original version: lines 232-234
This introduces the quote that comes immediately after the sentence.
In original version: lines 297-298
We feel that the sentence about ambiguous comments is an appropriate concluding sentence for that section.
In original version: lines 320-322
This introduces the quote that comes immediately after the sentence.
In original version: lines 461-462
These lines are in the discussion section anyway so we think the peer reviewer means 361-362? This sentence describes some of the findings about bisexual women.
Minor comments:
Line 64: the statement regarding the value of mixed methods compared to other types of studies should be backed up by a reference, unless this is an opinion of the authors, in which case this should be stated.
Response: Mixed methods systematic reviews have been around for 10 or more years but not many have been done so far. We have reworked the sentences in the background about mixed methods systematic reviews.
Line 73: in reference to unpublished research, the authors may wish to state specifically where they obtained this unpublished research, and to which studies they are referring.
Response:
We have explained what we mean by unpublished literature by inserting a sentence into the methods section “(ie grey literature reports available on LGBT organisation websites”. You can tell from the references which are grey literature reports, for example: Guasp, A. (2011). Lesbian gay and bisexual people in later life. Stonewall, London.
Section 2.1: the authors very completely state where they obtained the studies included in their manuscript. A chart or figure showing how many studies were obtained from each source evaluated would add to this section of the paper.
Response:
This is actually very hard to do as most of the studies were found in more than one database (or website in the case of grey literature). So the multiple sources, together with multiple references to some of the studies (ie 26 studies in 29 papers), and that some studies contributed to both qualitative and quantitative parts of the systematic review would give a rather confusing diagram which we feel would subtract rather than add to the reader’s understanding of this section.
2.3: I appreciate the explanation for why meta-analysis was not the chosen form of analysis. Lines 128 and 139 state that 22 and 23 qualitative studies, respectively were analyzed. Elsewhere in the paper 22 studies are sited; correction of this discrepancy is needed.
Response:
Thank you very much for pointing this out. It was 22 studies and the number has been corrected now.
Line 239 uses two separate tenses: chose and is. Consolidating the tense may be useful for readers to more clearly read this section.
Response:
Yes we agree that this was confusing. This and the next couple of sentences should have been in the past tense and have been corrected now.
3.1.8: the reasoning given for why bisexual and trans women were analyzed separately was useful; I appreciate the clarification.
Response:
Thank you.
Line 529: the strength mentioned is a strength of the data, rather than a strength of the review itself. This should be mentioned. Rather than having two separate strengths and weaknesses sections, it would make the manuscript more streamlined to combine these into one section.
Response:
Thank you for pointing this out – we have changed this sentence to show that it is the demonstration of consistency that is a strength of the systematic review and have moved it to the first paragraph of the discussion.
We realised the ‘Strengths and weakness in relation to other research’ heading was poorly worded and have changed it to ‘Comparison to previous research’ to make it clearer why this section is distinct from the ‘Strengths and weakness of the systematic review section’.